# PPMaP: Reproducible and Extensible Open-Source Software for Plant Phenological Phase Duration Prediction and Mapping in Sub-Saharan Africa

**Henri E. Z. Tonnang [1,2,*], Ritter A. Guimapi [3], Anani Y. Bruce [1], Dan Makumbi [1], Bester T. Mudereri [2,4], Tesfaye Balemi [5] and Peter Craufurd [1]**

1   International Maize and Wheat Improvement Center (CIMMYT), off UN Avenue, Gigiri, ICRAF House, Nairobi P.O. Box 1041-0062, Kenya; a.bruce@cgiar.org (A.Y.B.); d.makumbi@cgiar.org (D.M.); p.craufurd@cgiar.org (P.C.)
2   International Centre of Insect Physiology and Ecology (icipe), Nairobi P.O. Box 30772-00100, Kenya; bmudereri@icipe.org
3   Biotechnology and Plant Health Division, Norwegian Institute of Bioeconomy Research (NIBIO), P.O. Box 115, 1433 Ås, Norway; ritter.guimapi@nibio.no
4   Department of Animal and Wildlife Science, Midlands State University, P. Bag., Gweru 9055, Zimbabwe
5   Ethiopian Institute of Agricultural Research, Addis Ababa P.O. Box 2003, Ethiopia; tesfayeb2005@yahoo.co.uk
*   Correspondence: htonnang@icipe.org

**Abstract:** Understanding the detailed timing of crop phenology and their variability enhances grain yield and quality by providing precise scheduling of irrigation, fertilization, and crop protection mechanisms. Advances in information and communication technology (ICT) provide a unique opportunity to develop agriculture-related tools that enhance wall-to-wall upscaling of data outputs from point-location data to wide-area spatial scales. Because of the heterogeneity of the worldwide agro-ecological zones where crops are cultivated, it is unproductive to perform plant phenology research without providing means to upscale results to landscape-level while safeguarding field-scale relevance. This paper presents an advanced, reproducible, and open-source software for plant phenology prediction and mapping (PPMaP) that inputs data obtained from multi-location field experiments to derive models for any crop variety. This information can then be applied consecutively at a localized grid within a spatial framework to produce plant phenology predictions at the landscape level. This software runs on the 'Windows' platform and supports the development of process-oriented and temperature-driven plant phenology models by intuitively and interactively leading the user through a step-by-step progression to the production of spatial maps for any region of interest in sub-Saharan Africa. Maize (*Zea mays* L.) was used to demonstrate the robustness, versatility, and high computing efficiency of the resulting modeling outputs of the PPMaP. The framework was implemented in R, providing a flexible and easy-to-use GUI interface. Since this allows for appropriate scaling to the larger spatial domain, the software can effectively be used to determine the spatially explicit length of growing period (LGP) of any variety.

**Keywords:** plant development rate; temperature-dependent; landscape; multi-location trials

## 1. Introduction

The recent rise and evolution of data analytics have greatly modernized the worldwide agriculture environment thereby improving the efficiency of monitoring farming environments [1,2]. Mathematical and statistical modeling tools embedded within computer programs and packages have been used

intensively over the last decade with the intent to enhance agricultural production at scale [3–8]. The output of such technologies has revolutionized the empirical optimization of production and accurate predictions of output which has aided in precision agriculture planning and management [7,9,10].

This paper contributes a reproducible and practical software capable of providing plant phenological phase period prediction and mapping (PPMaP). The PPMaP package, freely available at https://github.com/Atoundem/PPMaP, combines data obtained from multi-location field experiments for plant phenology to derive temperature-dependent models for any crop variety. This information is then applied consecutively at an individual grid-level within a spatial framework to produce predictions at scale [8]. Unlike the majority of the available plant analysis tools such as the North Carolina State University/Animal and Plant Health Inspection Service Plant Pest Forecasting System (NAPPFAST) [9] that uses the degree-day, this software supports the development of process-oriented temperature-driven plant phenology models and interactively leads the user through a step-by-step progression until the production of a spatial ASCII (American Standard Code for Information Interchange) file for any region of interest.

The modules with which the PPMaP software was developed were cognizant of the nexus between stages of plant development and the various variables that influence each stage. Plant development denotes the scheduling of occurrences in the life cycle of a plant that can be described as the increase in weight, volume, length, or area of some of the parts or the whole plant [11]. These plant growth and developmental processes are used to delineate the phenological stages of a crop. However, for modeling purposes, it is essential to separate the two main processes i.e., rate of biomass accumulation and the length of growth, as they are affected by different environmental variables even within intra-species [12,13]. On one hand, the rate of biomass accumulation is primarily determined by the amount of light captured by plants over a range of temperatures, whereas the length of growth for a specific variety is typically dependent on daily temperatures. It has been insinuated that the probable rate of biomass growth is comparatively constant over space and time when temperature values are within the range for plant growth, while the period of growth, changes in space and time [11,12].

Naturally, plants are considered as chemical "machines" that are sensitive to the temperature that characterizes the environment in which they are grown. In this context, earlier research has reported that each mechanism of plant development (enzyme reaction, metabolic sequence, and physiological process) is temperature-dependent [14–17]. Therefore, it follows that temperature determines the progression of processes to the next developmental stage and their respective rate of development [18]. Numerous relationships have been established to portray the way temperature influences plant development [19–21]. These comprise degree-days, day-degrees, heat units, heat sums, thermal units, and growing degree-days [8,16,17]. Depending on geographic location, there is a dissimilar weighting to the night and day periods for a specific crop, hence a variety might have different periods of growth as determined by these locational temperature variations. Additionally, this is also in consideration of the specific developmental characteristics of a crop, for instance, maize (*Zea mays* L.) which stops development when it reaches physiological maturity regardless of the success in development [22]. Therefore, the formulation of mathematical expressions to represent the development of plants and insects has evolved with emphasis placed on determining the lowest temperature at which development is zero (base temperature ($Tb$)) and the optimum temperature ($To$) above which development ceases to increase or begins to decline.

Motivated by this postulate, several models have been formulated towards enhancing understanding of these phenomena, for which each model has its strengths and weaknesses [13,19,20]. In general, relationships have been established between the time of development (duration between life-phases) of plant and temperature either empirically or through process-based methods. Process-based equations are formulated and parameterized with the biological knowledge of the plant, while empirical models only capture the trends of the data [20]. Despite the abundance of literature on the use of temperature-dependent models for the prediction of plant developmental phases [20,22,23], no simple tool exists to help develop and project the phenological phases of plants at the landscape

level. Besides, the majority of web services applications and analytics, which are currently available, are often tailored for the developed world with nothing to support the majority of farmers in developing countries [9].

For instance, the complex role of genetic factors in determining the performance of genotypes in different environments and selecting superior genotypes in target environments was demonstrated by Dia et al. (2016) [7]. In their study, the Genotype x Environment interaction (G × E) analysis was applied to describe the stability of genotypes and the response of crop in a multiplication context. This method helps to estimate the statistical parameters that guide the selection of genes that have more adapted environmental and agronomic traits [7]. Additionally, other several applications and online tools for plant image identification, analysis, and sharing of leaf images to interpret and predict potential yield have been proposed [10,24,25]. All these tools provide mechanisms to handle large datasets and further facilitate the distribution of information among the growing scientific community. Additional examples of efforts and software that were developed to improve agricultural modeling include the CN-CLASS [21] that was initially developed to study Carbon (C) stock in forest ecosystems but was later modified to link crop phenological development and C allocation during the growth of maize. Also, CropScape is a web service-based application for discovering and disseminating geospatial cropland data products for decision making [4]. Other software that includes the LEAF-E was developed to analyze plant leaf growth through function fitting, whereas TIPS is an automated computer system for processing image-based phenotyping of maize tassels [26]. However, the weakness of all these methods is that they generalize and facilitate the extrapolation of the potential areas for the optimum growth of the variety based on the known genotype, thus missing the fundamental location-specific parameters.

Therefore, we introduce new software that solves the limitations of the existing phenology determining platforms. The PPMaP is a framework that runs on the 'Windows' platform and enables multi-location field experiments for plant phenology to derive temperature-dependent models for any crop variety and then provides a grid specific recommendation domain for the variety. This enables the successful extrapolation of the length of growing periods (LGPs) for any varieties with high levels of accuracy with the lowest potential risk of failure [14]. This software also counters the traditional methods in which breeders find the best varieties by planting them in a diverse set of locations to measure performance over many seasons. The number of locations that these breeders can test the new varieties is limited and can cause uncertainty when trying to choose the best genotypes for farmers [27,28]. Thus, using PPMaP breeders can accurately predict the performance of each genotype in untested environments and therefore make better decisions on which genotypes to move forward and provide to farmers [14]. Ultimately this results in better decision making and increased crop production to help address food security threats at a global level. In this current study, the phenology model was developed for maize to demonstrate resulting modeling outputs using data collected in high maize producing regions in Ethiopia and Nigeria. In sub-Saharan Africa, maize stands out as an essential staple food, providing approximately 25% of total calories in the average diet of the majority of the population [29,30].

## 2. Materials and Methods

The software analytics used in the development of PPMaP was founded on the following four principles: (i) local, regional and global relevance, (ii) representativeness of the major agro-ecological zones within the region of interest, (iii) dependence on high attribute and smallest datasets, and (iv) adequate validation of the results with field trial results.

### 2.1. PPMaP input Datasets

#### 2.1.1. Plant Phenology Datasets

The cycle of a plant is split into different development stages or phases. Maize (crop used to test the software) growth was divided into the vegetative (VP) and the reproductive (RP) phases. The VP

commences at emergence, goes through the *nth* leaf, and ends with tasseling and flower [15]. The RP starts at flowering and is comprised of silking, blister, milk, dough, and dent stages, and stops at physiological maturity, which is reached when all kernels have maximum dry weight [31].

Input data into PPMaP for modeling the developmental rate of a plant were the duration of the individual phase of phenology. The data were collected from multi-locations characterized by a large temperature gradient to estimate how the duration of each phenological phase responds to diverse growing conditions. Main observations to be recorded include the sowing date, days to emergence, days to flowering, and days to physiological maturity. Days to emergence can be recorded by counting the number of maize plants daily while the emergence date is when 50% of the plants in the plot have emerged above the soil surface. A few days after the emergence of all plants, many plants can be randomly selected and tagged for use to record the duration of the VP and RP. The flowering date was recorded when 50% of tagged plants produced male or female flowers. Similarly, days to physiological maturity were recorded when 50% of tagged plants attain physiological maturity. A sample of how the data were organized before inputting into the software is shown in Figure 1.

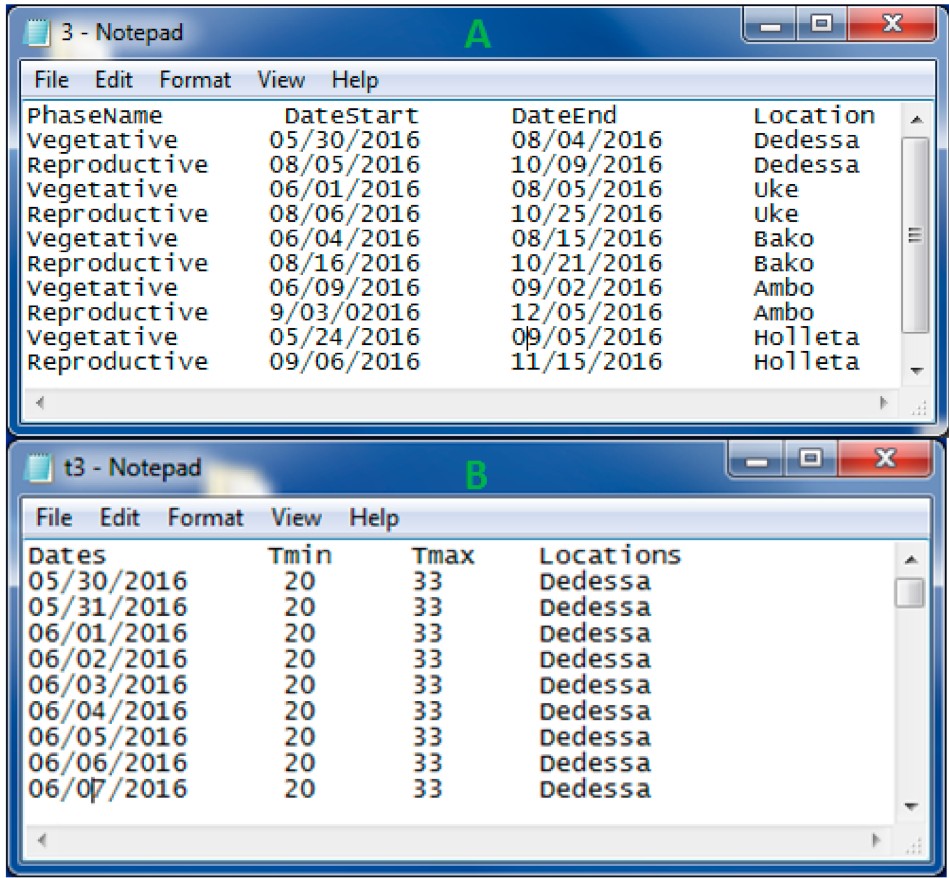

**Figure 1.** (A) Example of developmental phase start and end dates of a maize variety in five locations (Dedessa, Uke, Bako, Ambo, and Holleta) of Ethiopia, and (B) captures the average values of maximum and minimum temperatures in the five locations. Both files are saved as a txt-file (tab-delimited) and are used as input to plant phenology prediction and mapping (PPMaP) software. In (A), the first column indicates the developmental phase (vegetative or reproductive) of maize, the second column indicates the starting date of the phase, and the third column is the end date of the phase. Column four is the location in which the experimental trial was conducted. In (B), column one is the date of capture, column two, and three the minimum and maximum values of temperature respectively, while column four is the name of the location where the trails were conducted.

### 2.1.2. Temperature Datasets

The software uses two distinct sets of temperature. The first set of temperatures is collected from the weather stations within the vicinity of each field experiment. Minimum and maximum temperature for the trial period are required. The information is then processed to yield the average seasonal temperature for the site during crop growth. It is then associated with the duration of each plant phase when loading into the software (Figure 1). The second type of dataset needed by PPMaP for landscape mapping is the minimum and maximum weekly/monthly temperature organized in raster files with a *.flt* file format. The data can be obtained from any climate-based repositories and databases. To demonstrate the functionalities of the tool applicable to tropical maize, temperature data were extracted from the Worldclim website (www.worldclim.org). These datasets are presented in raster with "Float" file format (*.flt* and *.hdr* file) and consist of sets of climate layers (grids) with a spatial resolution ranging from 30 arc-seconds (~1 km) to 10 arc-minutes.

### 2.2. PPMaP Software Components and Data Processing Mechanisms

### 2.2.1. PPMaP Modeling Component

The modeling component was comprised of 82 factor-process-based development rate functions (Table S1) obtained from literature; which have been applied in agricultural production, either in the context of insect or crop phenology modeling [13,20,32]. A plant developmental rate model predicts the average proportion of development at a certain temperature or the fraction of development completed per unit time. These fractions were then accumulated under field conditions and were treated as independent variables during the factor-based modeling formulation. Model expressions in the PPMaP database were classified by the form of relationship and structure (linear, logistic, exponential, sigmoid, logarithmic, polynomial, and square root expressions).

Modeling with PPMaP starts with data standardization by dividing the development period of the individual phase by the mean development period of each field experimental site average temperature. This allows data obtained from multiple location trials with similar average temperature values to be merged for unique scrutiny. Because the direct selection of an appropriate mathematical expression to represent the development phase of a crop is challenging [13,20,33], the PPMaP software model component uses its in-built statistical criteria to compare, select, and display equations with the respective parameters and graph. Parameter values are estimated by fitting model equations to the field data. The Levenberg–Marquardt (LM) algorithm [34] was implemented in the software for the estimation of the model parameters. This algorithm combines both the 'steepest descent' and the Gauss–Newton method to appraise equations. It works iteratively to find the minimum of a function expressed as the sum of squares between the nonlinear model output and the observed datasets [34]. The PPMaP software offers an interactive process, in which initial parameter values for the selected model are altered; followed by the matching of model output to observed data. After the initialization of the models, the LM algorithm was launched and the goodness of fit procedure was used to find the parameters of the model. The model parameter comparison was done by the coefficient of determination adjusted r-squared ($R^2$_Adj) [13,20]; while the Akaike's Information Criterion (AIC) and the Model Selection Criterion (MSC) [20] were used to choose the model that best fits the input datasets (Figure 2).

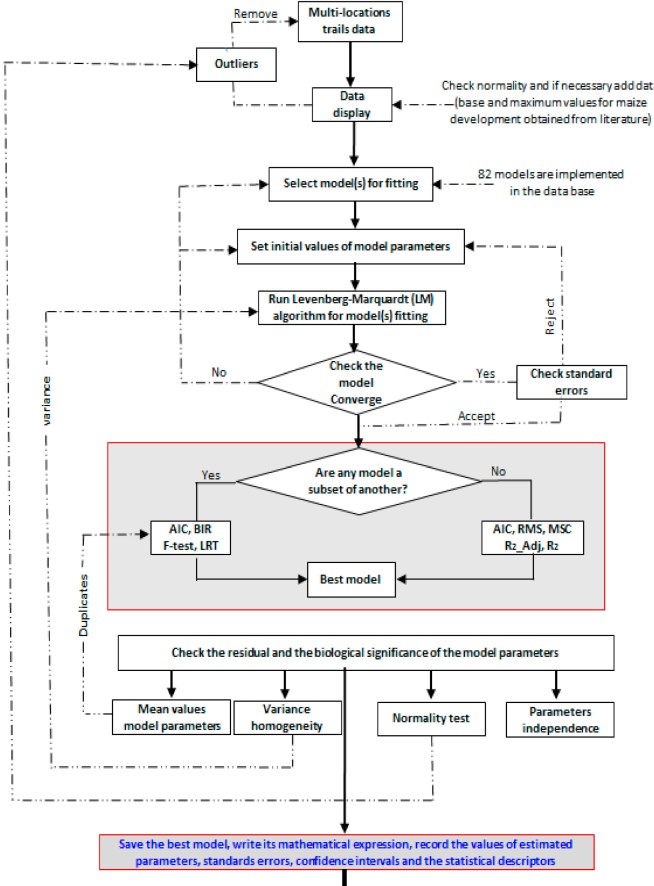

**Figure 2.** Flow chart demonstrating the steps within the PPMaP software from fitting plant phenology data obtained from multi-trails location to model selection and mapping. The thick, thin and dashed arrows are the essential steps, substeps, and feedback respectively. Akaike's Information Criterion (AIC), Model Selection Criterion (MSC), Likelihood Ratio Test (LRT), Bayesian Information Criterion (BIC) (Adapted from Archontoulis and Miguez, 2015).

### 2.2.2. PPMaP Mapping Component

A shapefile representing the boundaries of the region of interest is divided into grids and loaded into the software. The model obtained during the modeling step is run at each grid of the raster file. The gridded temperature data were also loaded and with an in-built function, the values were extracted from the database simultaneously and then arranged in a matrix format using longitude as a column and latitude as a row. A point object picks the model representing the plant development rate for a specific phase, and then sequentially replaces the variable temperature in each grid by the value obtained from the database. After computation in each grid, a new matrix was generated with the values of the development rate in the respective grid of the raster file. These values were then inverted (1/development rate) to estimate the number of days used to complete each phase of plant development. The results were converted into ASCII files and can be transferred to any geographical information system (GIS) [35] software for better visualization.

### 2.3. PPMaP Software Implementation

The following two programming methods were encompassed and combined to implement the PPMaP software: (i) a procedural approach that employed the R [36] constituent for model fitting and estimation of statistical criteria [37], and (ii) Java object-oriented programming language was used to interconnect the software components and develop the graphic user interface (GUI). This was

developed on the Eclipse platform [38]. The R constituent of the PPMaP comprises of R-scripts used to code mathematical expressions (models), processes for parameter estimation and model selection, and the mapping environment. Many R packages such as 'minpack.lm' [39], 'MASS' [40]; 'maptools' [41], and 'maps' [42] were employed to handle nonlinear regression analysis and model fitting and to carry out spatial operations leading to the generation of maps.

The Java object-oriented programming allows for the development of a rapid and reliable end-user interface with objects, subclasses of image, color, and font integrated into the Eclipse workbench. The 'Rserve' is a tool that is used to evaluate R codes and ensure their integration into the Java application [43]. The 'Rserve' was applied within the PPMaP software to ensure two-way communication between the R constituent and the Java object-oriented codes. When a request is made, 'Rserve' unlocks the link to collect details of the request, and then creates another link with the R constituent to transfer information for execution and sends back the result to the GUI for display. The GUI offers a direct connection between the user and the PPMaP software system. Numerous utilities were developed to bring PPMaP functionalities together in a stop screen layout to make it operate intuitively. The PPMaP software iteratively helps the user to create a project (Figure 3); import, modify, plot, and visualize data (Figure 4); carry out modeling (Figures 5 and 6), and to project the temperature-dependent model at scale (Figure 7).

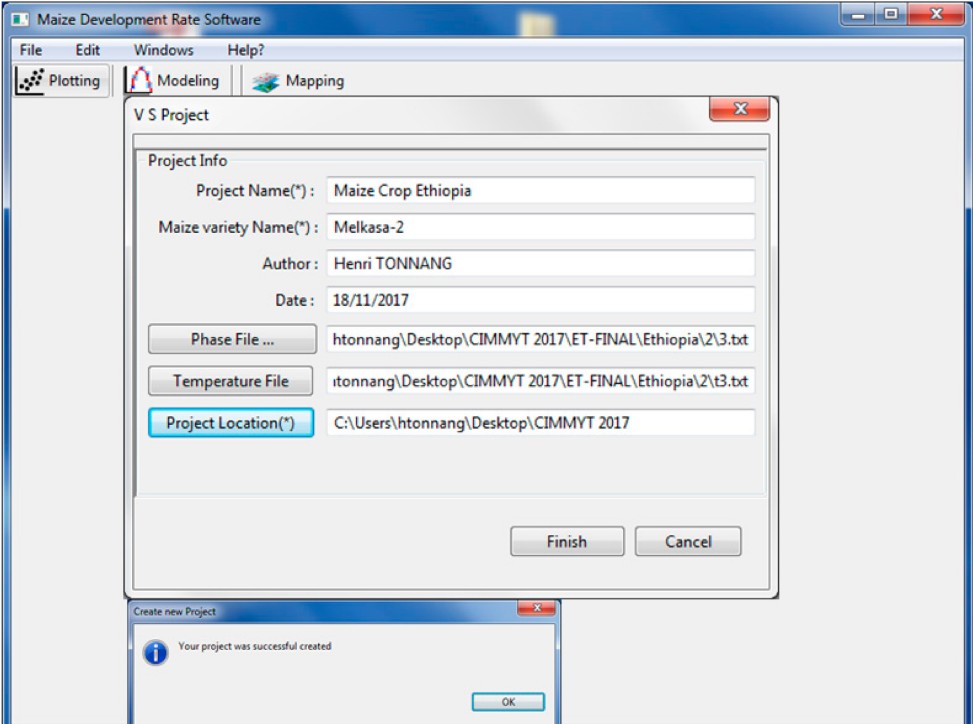

**Figure 3.** Display of the PPMaP software window for project creation. Under project info, there is the project name (Maize Crop Ethiopia), the maize variety name (Melkasa-2), the author (Henri TONNANG), and the date (18 December 2017) on which the project was created. Phase file designates the path to load (Figure 1). The temperature file is the path to load (Figure 1). Project location designates the folder in which all project files are stored. A well-created project prompts the software to display the message "*Your project was successfully created*".

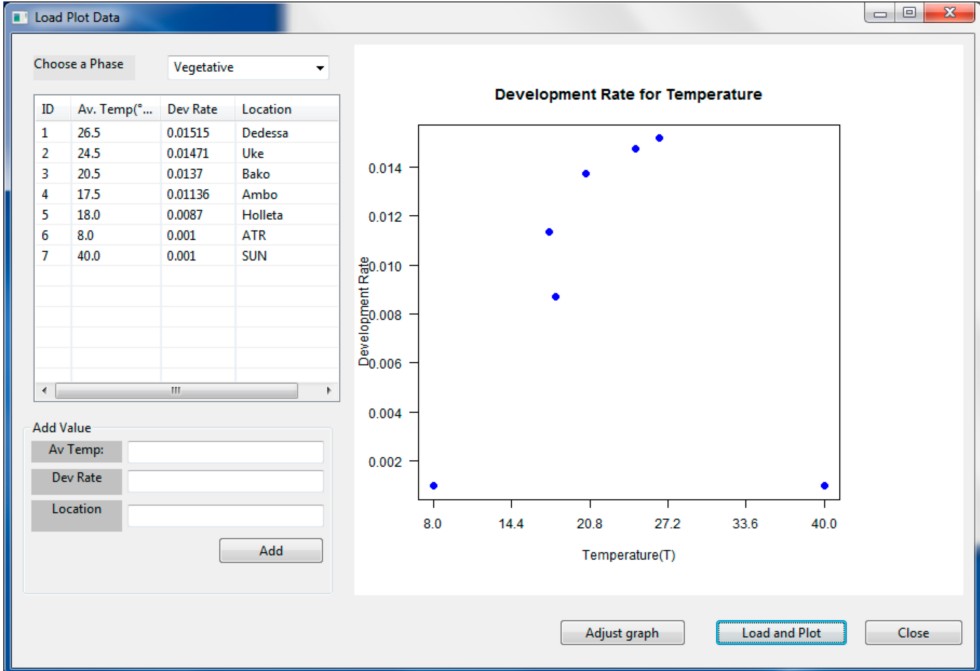

**Figure 4.** PPMaP data file display window. It contains the compilation of data of the vegetative phase of a maize variety. The columns on the left box contain, the ID, average temperature, numerical values of the development rate, and the locations. The right box shows in two dimensions the data recorded in the left box.

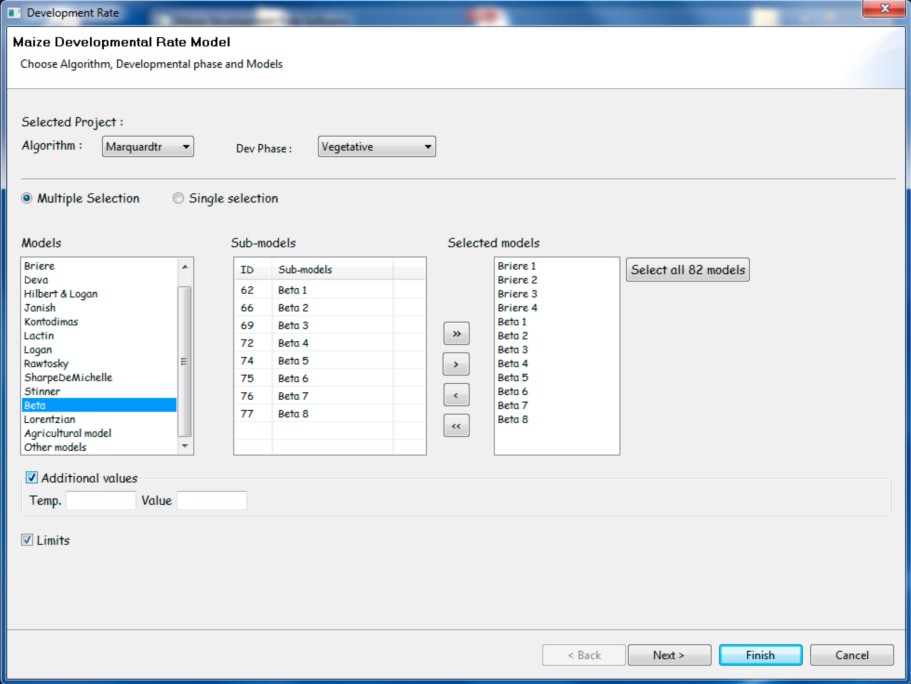

**Figure 5.** Illustrates the user interface of the "modeling component" of the PPMaP software. It shows the library of mathematical expressions that are used to describe the relationship between plant phase development and temperature. The left box contains the models characterized by the name of the author that pioneers the formulation. The center box contains the sub-models, which are derivative of the models from the left box. The right box contains the models selected for fitting. For example, in Briere$_i$, the index *i* represents the number of the derived model from the original Briere model.

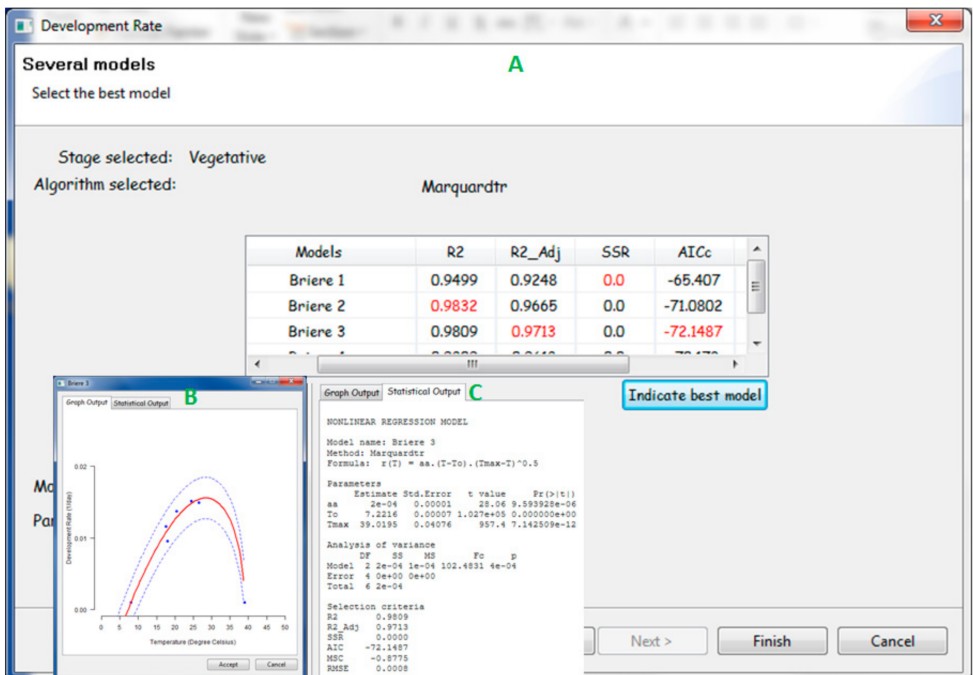

**Figure 6.** (**A**) Window of the PPMaP software presenting diverse models, each with corresponding model selection criteria for comparison; usually the best model has the smallest value of Akaike's Information Criterion (AIC). (**B**) Example of points of data (blue) for a variety of maize at the vegetative phase fitted by Briere temperature-dependent mathematical expression (red curve). (**C**) The statistical criteria for model selection, the model parameters, and the (ANOVA) analysis of variances are shown.

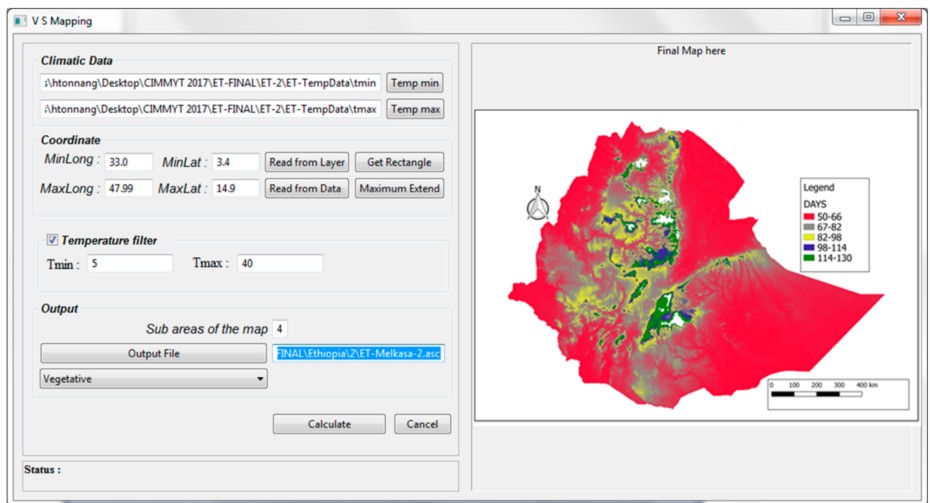

**Figure 7.** PPMaP window showing the mapping section of the software. The paths for inputting minimum and maximum temperatures are shown. Below these paths are the geographical coordinates of the region chosen for mapping. The map of Ethiopia showing the number of days a maize variety could spend during the vegetative phase appeared in a box at right. Depending on the location, this maize variety can spend from 50 to 130 days to start flowering.

### 2.4. PPMaP Model Validation and Performance at Scale

PPMaP software considers the validation as the level to which a selected model rightly predicts the duration of an individual phase of the plant when compared to experimental data. To carry out this procedure, independent data acquired from various field trial sites that were not used during model development and calibrations are recommended for use. The temperature at point location was

replaced in the expression of the inverse of development rate to estimate the duration of the individual phase of the plant, and the result was compared with actual recorded data from the field trial. A well-validated model was used to measure the performance at scale.

Figure 8 shows the phenology maps of two maize varieties (MV-1 and MV-2) that were grown both in Ethiopia and Nigeria, respectively. The period to complete the individual phase during the growing period in each country differs considerably due to variability in the temperatures. The growing period of variety MV-1 in Nigeria was predicted to be between 40 to 50 days, whereas in Ethiopia it could take from 50 to 130 days. MV-2 also presents a similar difference in the growing period in Nigeria (59 to 64 days) and Ethiopia (60 to 170 days). These results demonstrate the high level of variability that exists in agro-ecological areas in an individual country. It is noted that both maize varieties (MV-1 and MV-2) can successfully grow in most parts of Nigeria and Ethiopia. However, the time each variety will take to reach maturity will differ from one location to another. With Ethiopia having wider variability in altitude compared to Nigeria, this translates to diverse temperature regimes and the time taken by both maize varieties to reach maturity is longer than in Nigeria. At higher altitudes, the temperatures are lower, making the duration of development longer than at lower altitudes.

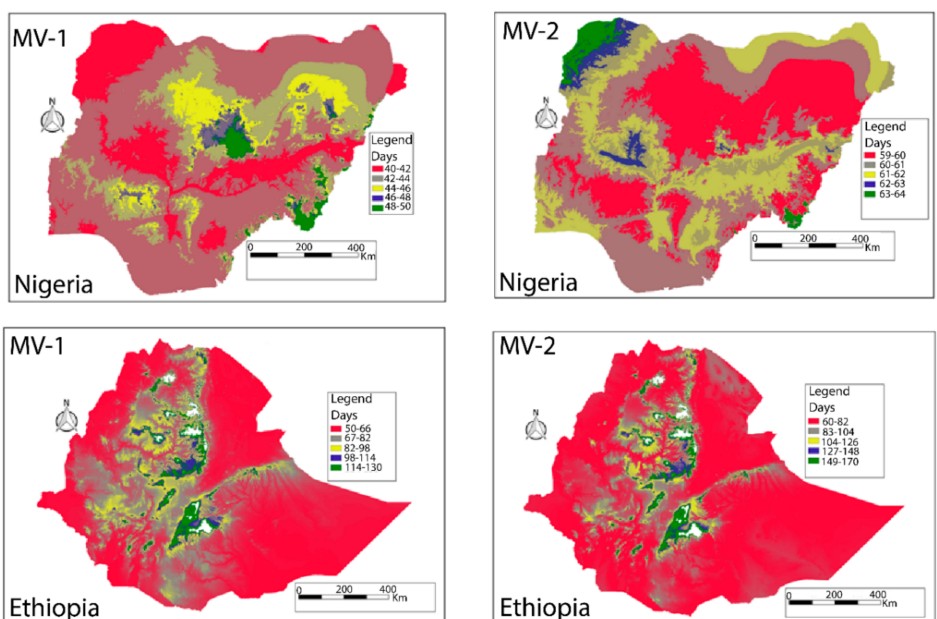

**Figure 8.** Maps of Nigeria and Ethiopia showing the number of days for two maize varieties (MV-1 and MV-2) would take during each phase (vegetative or reproductive) of development in different agro-ecologies. The small white dots are areas with missing values of temperature.

## 3. PPMaP Software Usefulness and Discussion

Phenology is an area of research that assesses the cyclical recurrence of events in the growth and development of plants and insects [12]. Thus, predicting plant phenology is important as it can be used to improve crop productivity. Precisely, forecasting stages of crop development duration are needed to find genotypes with the desired growth period that allows growers to optimize yields. Adequate forecast of plant phenological events further guides in management decision making like the timing of pesticide application, scheduling the timely harvest of crops, or synchronizing the flowering of cross-pollination crops for seed production [11].

Notwithstanding the multitude of literature [11,12,20] on the use of temperature-dependent models for modeling plant developmental phases, no software is available to predict plant phenological events. The PPMaP software presented here is a unique computer-aided tool capable of analyzing multi-location field trial information of plants to yield complete phenology models. PPMaP encompasses a full library of mathematical expressions and statistical criteria that guide users to discovery and to select

the best model to describe the temperature dependence of the development for each phase of the plant. The starting point in developing plant phenology models with PPMaP is data standardization and grouping according to stage and site-specific characterized by the average temperature in which the trials were conducted. Fitting the developmental rate curves directly to the reciprocal of developmental duration follows this step. To account for variability among plants, PPMaP applies the distributional concept demonstrated in [44,45], which assumes that variation in developmental rates at constant temperature is the direct result of variability in the concentration of rate-controlling enzymes while the developmental rate variance is proportional to the mean developmental rate.

An important attribute of PPMaP is the relative ease and speed with which the overall process of producing the model and projecting at scale to yield the phenology map of the crop is done. The software environment that runs on the 'Windows' platform makes programming and coding unnecessary and hence allows for the estimates of the duration of plant phases effortlessly available to researchers. PPMaP has a graphical user interface and a collection of appropriate models for representing different processes that determine the plant life cycle. Selection is based on analytic and adequate evaluation criteria. Fitting mathematical expressions using either maximum likelihood estimation (MLE) or least-squares estimation (LSE) and treatment medians is difficult and sometimes often leads to over parameterization and convergence [20]. Therefore, PPMaP offers a solution to the problem whereby a user may be requested to increase the number of data points or use the in-built software features that allow the use of censored data to facilitate the analysis. In some cases, model convergence with certain initial parameters may remain an issue. However, PPMaP provides an option to automatically modify the initial parameters of models to ease the convergence of the fitting algorithm. However, fitting mathematical expressions remains a complicated task and should be done carefully. There is no standard method to choose among competing equations; that is, the decision on choosing a model should not be left to a computer program, and thus the user of PPMaP needs to critically consider and identify an appropriate equation that best delivers biologically meaningful and statistically significant parameters suited to their objective and species or variety [20].

In Rykiel, (1996), validation is defined as the degree to which a model output matches independent data sets. Thus, PPMaP uses the same concept and has a mechanism for using independent datasets (not used for model development and calibration) obtained from diverse field trials and sites to make predictions and compare results. In this context, validation is an important instrument for scaling life cycle events of plants over a wide spatial and temporal range and for examining the crop reactions to changing or novel climatic conditions, and it provides some level of understanding and adaptation in different agro-ecological zones.

Taking crop research to scale requires a change in the way studies are conducted as well as the extent to which essential advice to farmers can be provided. Two types of frameworks exist the top-down approach consisting of analyzing grid cells over a wide area and then downscaling the results to a single cell, while the bottom-up approach involves analysis from localized information of a cell followed by a spatial upscale [46]. Consequently, PPMaP software supports the concept of researching to the appropriate scale by using the bottom-up approach. Inputting phenological data of plants to carry out analysis with geospatial representation is thus a critical step to tailoring solutions at the landscape level. Considering the diversity of agro-ecological zones where crops are cultivated, we argue that it is unproductive to conduct research studies on plant phenology without providing means of upscaling results to landscape-level while still safeguarding field relevance. Herein, we presented software embedded with a method that allows appropriate scaling to larger spatial domain research findings for decision-making. Global warming might be affecting the phenology of plants in their respective natural environments.

Tools based on hyperspectral imaging exist for differentiating crop phenology and seed varieties [10,24]. Plant phenotyping [10,27] is an emerging field of research that links genomics with plant ecophysiology and agronomy. It demonstrates that the functional plant body (phenotype) is produced during plant growth and development from the dynamic interaction between the genetic

setting (genotype) and the physical area in which plants develop (environment). The understanding of the connections and feedbacks mechanisms existing between phenotype, genotype, and the environment is paramount for increasing crop productivity, therefore, we hope that PPMaP, which helps to understand how genotype can respond to environmental conditions, will contribute towards improving knowledge on this topic.

In several countries of sub-Saharan Africa (SSA), information on crop seed packages is often incomplete; crop varieties are classified by altitude (low, medium, and high) and maturity groups (early, intermediate, and late). PPMaP is a tool that could help provide location specific and near accurate length of crop growth duration of different varieties. Consequently, we propose to compact the phenology maps of different maize varieties produced by PPMaP, add features such as the name, type, color, potential yield, maturity class, ecology, resistance, and tolerance level to pests/diseases for individual varieties to develop applications. This software can be used to enhance the capacity of extension and agribusiness service providers (agro-dealers) in providing recommendations at various scales especially in countries where the landscape varies considerably in altitude such as Ethiopia. When a new variety is developed, the norm is to conduct multi-environmental trials to evaluate the performance of the variety before release. Such trials are very expensive and PPMaP can be a starting point for reducing the number of experimental sites and thus associated costs.

## 4. Conclusions

This study has demonstrated that PPMaP is a tool that has vast potential to provide a location-specific and near accurate length of crop growth duration of different varieties by supporting the concept of researching to the appropriate scale. It is an object-oriented reproducible and extensible framework for phenology mapping at scale. Besides, PPMaP provides an easy-to-use comprehensive framework to perform the entire modeling process without the extensive need for programming skills hence user friendly to potential users of any computer skills level. The software is thus designed to enable users to extend it and share the new data, methods, or procedures to reproduce them by other users.

**Supplementary Materials:** The following are available online at http://www.mdpi.com/2077-0472/10/11/515/s1, Table S1: Models in PPMaP software.

**Author Contributions:** Conceptualization, H.E.Z.T., R.A.G., and P.C.; methodology, H.E.Z.T. and R.A.G.; software, H.E.Z.T.; validation, H.E.Z.T., A.Y.B., D.M., and T.B.; formal analysis, H.E.Z.T. and R.A.G.; investigation, H.E.Z.T., T.B., and P.C.; resources, P.C.; data curation, H.E.Z.T., and T.B.; writing—original draft preparation, H.E.Z.T., R.A.G., B.T.M., A.Y.B., and D.M.; writing—review and editing, H.E.Z.T., A.Y.B., B.T.M., R.A.G., P.C., and T.B.; visualization, H.E.Z.T. and T.B.; supervision, D.M. and P.C.; project administration, P.C.; funding acquisition, P.C. All authors have read and agreed to the published version of the manuscript.

**Funding:** The present study was executed by the International Maize and Wheat Improvement Centre (CIMMYT) and International Institute of Tropical Agriculture (IITA) as part of the TAMASA (Taking Maize Agronomy to Scale in Africa) project, made possible by the generous support of the Bill and Melinda Gates Foundation (contract OPP1113374). Any opinions, findings, conclusions, or recommendations expressed in this publication are those of the authors and do not necessarily reflect the view of the donor.

**Conflicts of Interest:** The authors declare no conflict of interest.

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
