# Peer review of "PPMaP: Reproducible and Extensible Open-Source Software for Plant Phenological Phase Duration Prediction and Mapping in Sub-Saharan Africa"

_agriculture, doi:10.3390/agriculture10110515_

Round 1
Reviewer 1 Report
The topic of this paper is interesting and topical. It is well written. However there is one big question before it can be considered to publication.
The paper is entitled as extensible and throughout the article authors suggest it can and should be used troughout different climatic regions as well as for different crops. However the article itself only treats sub-Saharan Africa and one crop - maize. Either the title and description should reflect that or additional information (or references) is needed about the possibilities, challenges and specificities about using the same concept under different growing conditions and different crops. For instance, how does your model account fo late/early nightfrost or very variable daylengths across the climates, enhancing development in nordic regions?
Also, a sincere quetsion of interest and since you emphatize the simplicity of the software for users and non-modellers - have you tested it out on real end-users (by which I mean not scientists, but farmers)? It would be really interesting to include tje real-life experience.
Thank you.
Author Response
Reviewer #1
The topic of this paper is interesting and topical. It is well written.
Thank you for your positive comment
However, there is one big question before it can be considered to publication.
The paper is entitled as extensible and throughout the article authors suggest it can and should be used troughout different climatic regions as well as for different crops. However, the article itself only treats sub-Saharan Africa and one crop - maize. Either the title and description should reflect that or additional information (or references) is needed about the possibilities, challenges and specificities about using the same concept under different growing conditions and different crops. For instance, how does your model account fo late/early nightfrost or very variable daylengths across the climates, enhancing development in nordic regions?
Thank you again for this comment. Indeed, as noted by the reviewer our software is applicable within tropical climates where the length of day and night are almost equal. Therefore, we have altered our title to show that in its current, i.e. state presented in this manuscript, it applies to sub-Saharan Africa where it has been tested successfully. However, it is extendable and can be further modified to fit other regions.
Kindly see line 4 and 34
Also, a sincere question of interest and since you emphatize the simplicity of the software for users and non-modellers - have you tested it out on real end-users (by which I mean not scientists, but farmers)? It would be really interesting to include tje real-life experience.
We took a kind note of the reviewer’s comment. As highlighted in the manuscript, the software is simple to breeders and other scientists who have some level of computer skills however not to the level of the common farmer. We have however simplified this software further by providing a mobile application known as the Maize Variety Selector (not presented in this current manuscript) that is more user-friendly to the farmer. The work on the Maize Variety Selector is already ready for submission and we will be glad to share once it is published.
We have also motivated in the introduction and have demonstrated that this software can be used for decision making by breeders and scientists to help farmers. Please kindly see lines 126-131 which read “The number of locations that these breeders can test the new varieties is however limited and can cause uncertainty when trying to choose the best genotypes for farmers [27,28]. Thus, using PPMaP breeders can accurately predict the performance of each genotype in untested environments and therefore make better decisions on which genotypes to move forward and provide to farmers [14]. Ultimately this results in better decision making and increased crop production to help address food security threats at a global level”.

Reviewer 2 Report
In this study the authors develop an open-source software to predict and map plant phenological phase duration. In this reviewer’s opinion, this manuscript was well written and is worthy of considering to publication in agriculture with two minor changes. (1) To add a flowchart explaining input data, modelling components, mapping component and outputs will facilitate the reader to follow and understand PPMaP software more easily. (2) I found difficult to follow the bottom-up approach of the software, I suggest a better explanation.
Author Response
Reviewer #2
In this study the authors develop an open-source software to predict and map plant phenological phase duration. In this reviewer’s opinion, this manuscript was well written and is worthy of considering to publication in agriculture with two minor changes.
Thank you very much for your encouraging comment.
(1) To add a flowchart explaining input data, modelling components, mapping component and outputs will facilitate the reader to follow and understand PPMaP software more easily.
We provided a flowchart presented in Figure 2 in the manuscript.
Kindly see lines 209-214
(2) I found difficult to follow the bottom-up approach of the software, I suggest a better explanation.
Two types of frameworks exist the top-down approach consisting of analyzing grid cells over a wide area and then downscaling the results to a single cell, while the bottom-up approach involves analysis from localized information of a cell followed by a spatial upscale. The training of the prediction is conducted by data that is collected at scale (smallest pixel size resolution to map) and then used to conduct predictions for the entire country for instance. A sample study using this approach and software was conducted by Tonnang et. al. (2018)
Tonnang HEZ, Makumbi D, Craufurd P (2018) Methodological approach for predicting and mapping the phenological adaptation of tropical maize (Zea mays L.) using multi-environment trials. Plant Methods 14:1–12. https://doi.org/10.1186/s13007-018-0375-7

Reviewer 3 Report
The manuscript entitled "PPMaP: reproducible and extensible open-source software for plant phenological phase duration prediction and mapping" is a software article whose main objective is to provide a tool capable of spatial mapping of any region of interest for temperature-dependent plant phenology models. The software could be a useful tool to help breeders around the world make better decisions and increase crop production. However, I still have some major comments:
- Can the software be installed on any Operative System (Linux, Mac, Windows) ? I tried to install it and I couldn't.
- Although I couldn't follow the example provided in the user guide, I think it could be improved, indicating not only how to use the software with the provided "toy" dataset, but also with the data provided by the user. I think it is a key piece of the paper and works as the supplementary material.
- The authors claim the software can be applied to any crops data, however they only show a single example with maize. If more data are available, it could improve the software visibility and usability.
- I think that a R package (without using the java environment) also can help to increase the software usability.
- I am aware that it could take some extra work, but a web platform with the software can also help to using it.
- It would be useful for end users if the software could seek out and download the temperature data automatically.
Minor:
- The quality of the figures must to be improved. For instance in the figure 8, i can't read the legend.
- Line 167, can you explain more detailed how the temperatures are directly obtained?
- Regarding the 82 models from the literature, can the authors explain whether any one of them is better than others or which one they would generally recommend?
- The authors mention that one of the main advantages of this software is their speed, can you give more details about that? How long does it take to run the whole process on a specific computer?
- how many observations the input file can have?
- This expression could be simplified Notwithstanding the multitude of literature on the use of temperature-dependent models for modelling plant developmental phases, no software can help non-modellers to establish mathematical expressions for useful prediction of plant phenological events.
Notwithstanding the multitude of literature on the use of temperature-dependent models for modelling plant developmental phases, no software is available.
Author Response
Reviewer #3
The manuscript entitled "PPMaP: reproducible and extensible open-source software for plant phenological phase duration prediction and mapping" is a software article whose main objective is to provide a tool capable of spatial mapping of any region of interest for temperature-dependent plant phenology models. The software could be a useful tool to help breeders around the world make better decisions and increase crop production. However, I still have some major comments:
1. Can the software be installed on any Operative System (Linux, Mac, Windows) ? I tried to install it and I couldn't.
The software can be installed on the Windows platform
2. Although I couldn't follow the example provided in the user guide, I think it could be improved, indicating not only how to use the software with the provided "toy" dataset, but also with the data provided by the user. I think it is a key piece of the paper and works as the supplementary material.
Thank you for this comment that seeks to improve the ease of use of our software. To that effect, we have provided a step-by-step procedure that the reader can follow using their data as suggested by the reviewer. We have referred readers to a paper that also provides a good example of the use of the software. Please refer to Tonnang et al. (2018). This paper evaluates 82 different temperature-dependent models to predict the phenological development of 22 diverse tropical maize varieties using existing multi-environment trial data.
Tonnang HEZ, Makumbi D, Craufurd P (2018) Methodological approach for predicting and mapping the phenological adaptation of tropical maize (Zea mays L.) using multi-environment trials. Plant Methods 14:1–12. https://doi.org/10.1186/s13007-018-0375-7
3. The authors claim the software can be applied to any crops data, however they only show a single example with maize. If more data are available, it could improve the software visibility and usability.
Thank you for this observation, we use maize as a flagship crop considering its importance in sub-Saharan Africa. Indeed, the software can be used for any crop that has growth phases that can be subdivided into growth days, hence we used maize as a demonstration crop. Considering the resources required to collect sample data at scale, demonstrating another crop would be possible in another study. However, the principles and data required to run the software can be implemented for any crop as we have rightly claimed.
4. I think that a R package (without using the java environment) also can help to increase the software usability.
Thank you for this very critical and important suggestion for future work, indeed we agree, and this can be done as separate work from the work we present here in this current manuscript. We look forward to taking it up and operationalize this suggestion.
5. I am aware that it could take some extra work, but a web platform with the software can also help to using it.
Thank you again for this insightful suggestion for future work, indeed we agree, and this can be done as separate work from the work we present here in this current manuscript. We look forward to taking it up and operationalize this suggestion.
6. It would be useful for end users if the software could seek out and download the temperature data automatically.
Thank you again for this insightful suggestion for future work, indeed we agree, and this can be done as separate work from the work we present here in this current manuscript. We look forward to taking it up and operationalize this suggestion.
Minor:
1. The quality of the figures must to be improved. For instance in the figure 8, i can't read the legend.
The figure has been improved thank you for your suggestion
Kindly, please see lines 303 and 304
2. Line 167, can you explain more detailed how the temperatures are directly obtained?
We collected data from weather stations within the vicinity of the sample crop fields that were used in this study.
Kindly see lines 170-171
3. Regarding the 82 models from the literature, can the authors explain whether any one of them is better than others or which one they would generally recommend?
The selection of the models from the list of 82 (Table S1) depends on the input data. Therefore, there is no absolute best model. A detailed description of the models is also provided in Tonnang et.al. 2018, where the use of sample data is precisely described. In the referred paper, the authors evaluate the 82 different temperature-dependent models to predict the phenological development of 22 diverse tropical maize varieties using existing multi-environment trial data. The software further include statistical criteria for model selection
Tonnang HEZ, Makumbi D, Craufurd P (2018) Methodological approach for predicting and mapping the phenological adaptation of tropical maize (Zea mays L.) using multi-environment trials. Plant Methods 14:1–12. https://doi.org/10.1186/s13007-018-0375-7
4. The authors mention that one of the main advantages of this software is their speed, can you give more details about that? How long does it take to run the whole process on a specific computer?
We think that the speed with which the whole process takes depends on the computer RAM, the extent of the area under study (region of interest), and the spatial resolution being used. This information may be biased as brands and computer specifications greatly influence the speed of processing. However, in the coding of the software, we have employed an advanced computing technology that allows the process to be computed simultaneously instead on grid after grid. While developing maize phenological prediction maps for Kenya, the first version of the software was taking days, it is currently a matter of hours.
5. how many observations the input file can have?
To produce a graph as shown in Figure 6, at least 4 observations are necessary. Thank you
6. This expression could be simplified Notwithstanding the multitude of literature on the use of temperature-dependent models for modelling plant developmental phases, no software can help non-modellers to establish mathematical expressions for useful prediction of plant phenological events.
We took a kind note of the reviewer’s comment and the sentence now reads “Notwithstanding the multitude of literature on the use of temperature-dependent models for modeling plant developmental phases, no software is available to predict plant phenological events.”
Round 2
Reviewer 3 Report
The manuscript should specify that the software can only be installed in Windows platform.
The installation specifications should be included in the manual, including the platform on which the software operates.
Author Response
Reviewer comment #1
- The manuscript should specify that the software can only be installed in Windows platform.
Response#1
Thank you for highlighting to us this critical oversight. We have added this information highlighted in red color therein in the Abstract (Kindly see line 31) This software runs on the ‘Windows’ platform and supports the development of process-oriented and temperature-driven plant phenology models by intuitively and interactively leading the user through a step-by-step progression to the production of spatial maps for any region of interest in sub-Saharan Africa;
in the Introduction (Kindly see line 120) The PPMaP is a framework that runs on the ‘Windows’ platform and enables multi-location field experiments for plant phenology to derive temperature-dependent models for any crop variety and then provide a grid specific recommendation domain for the variety.;
and in the Discussion (Kindly see line 332) The software environment that runs on the ‘Windows’ platform makes programming and coding unnecessary, and hence allows the estimates of the duration of plant phases effortlessly available to researchers.
of the revised version of our manuscript.
Reviewer comment #2
- The installation specifications should be included in the manual, including the platform on which the software operates.
Response#2
Thank you once again for this important suggestion. We have already started to work on adding this clarification on our online manual as suggested by the reviewer.